# Reimplementing the Adversarially Reweighted Learning model by Lahoti et al. (2020) to improve fairness without demographics

**J. Mohazzab, C.A. Wortmann, L.R. Weytingh, and B. Brocades Zaalberg**
Faculty of Science
University of Amsterdam

## Reproducibility Summary

### Scope of Reproducibility

It is often the case in Machine Learning systems that the used data does not contain protected group membership due to privacy rules and regulations. This makes it difficult to improve fairness for disadvantaged subgroups. As a solution, Lahoti et al. propose Adversarially Reweighted Learning (ARL) [1]. They claim that ARL significantly improves fairness for computationally identifiable subgroups.

### Methodology

In this project we aimed to reproduce, replicate, and evaluate the results presented by Lahoti et al. First, the open-source TensorFlow implementation of the ARL model was used to test the reproducibility of the results. Second, the ARL model was re-implemented in PyTorch to test the replicability of the results. Finally, the significance of the ARL model was tested against a baseline model using P-value tests. We trained and evaluated the models in about half a minute per model iteration (i.e. for fully training and evaluating the model) on a 2,3 GHz 8-Core Intel Core i9 processor.

### Results

Our findings suggest that (1) the paper is not reproducibile, (2) the paper is replicable, yet (3) the results are not significant. The main results were reproduced within 2% of the reported values. However, with limited knowledge of the original hyperparameters used and the inability to produce several additional metrics presented in the paper we concluded the paper to not be reproducible. The PyTorch implementation produced results within 1% of the reported values, suggesting that the paper is replicable. However, the results proved to be insignificant when compared to a baseline model.

### What was easy

The paper by Lahoti et al. was concise and clearly structured. This, in combination with the well documented open-source TensorFlow implementation, provided us with clear guidance when re-implementing the ARL model in PyTorch.

### What was difficult

Pre-processing the data proved difficult. In addition, some details regarding the model were not mentioned in the paper. Therefore, we had to make some impactful assumptions about e.g. the amount of training steps, and the original hyperparameters used.

### Communication with original authors

The authors were contacted by email about some missing details in their paper. However, we did not receive a response.

34rd Conference on Neural Information Processing Systems (NeurIPS 2020), Vancouver, Canada.

# 1 Introduction

The integration of Machine Learning (ML) in decision making models has significantly increased in recent years. In many of these models the decisions have important consequences for the involved individuals. Therefore, it is of high importance that these models do not exhibit discriminatory behavior. That is, the use of personal data and novel technologies should not cause new inequalities or enhance existing ones [2]. However, research has shown that many ML systems amplify disparities in classification accuracy between protected subgroups [3, 4, 5].

To prevent these inequalities it is important to consider fairness in ML systems. Fairness is defined as the absence of any prejudice or favoritism towards an individual or a group based on their inherent or acquired characteristics [6].

Literature on ML fairness often assumes that protected features such as race and gender are present in the dataset. However, in practice these features are often not included because of privacy rules and regulations [7, 8].

In the paper "fairness without Demographics through Adversarially Reweighted Learning" [1], Lahoti et al. ask the main research question: "How can we train a ML model to improve fairness when we do not know the protected group memberships?".

To answer this question, the assumption is made that subgroups within the data are computationally-identifiable through other attributes. Based on this assumption, Lahoti et al. propose the Adversarially Reweighted Learning (ARL) model. The ARL model aims to maximize the minimum utility for worst-off groups. As such, the notion of fairness used in this paper is the Rawlsian principle of Max-Min welfare for distributive justice [9]. The proposed ARL model combines *re-weighting* [10, 11, 12], *adversarial learning* [13, 14, 15, 16] and *focal loss* [17]. This method is an in-processing approach for improving fairness, in which the objective of the algorithm is alternated during training to emphasize fairness.

The purpose of this report is threefold: (1) to test the reproducibility of the paper. This entails reproducing the results reported in the paper using the original TensorFlow implementation by Lahoti et al. [1]. (2) To test the replicability of the paper. This entails re-implementing the ARL model in PyTorch [18]. The paper by Lahoti et al. does not include tests for validating the significance of the presented contributions. Therefore, (3) we apply a significance test to the obtained results.

# 2 Background

**Rawlsian Max-Min fairness**    In ML literature, multiple fairness notions have been suggested. They can be organized into three groups: individual fairness [19], group fairness [20] and fairness notions that aim to improve per-group performance [21]. In the paper by Lahoti et al., the per-group notion of fairness is used.

In order to improve per-group fairness, a criterion is needed to determine how the utilities of individuals in a group should be distributed. Well known notions of per-group fairness are Utilitarian fairness, Pareto fairness and Rawlsian Max-Min fairness [21]. Following the article of Lahoti et al., the Rawlsian Max-Min fairness notion will be used. In the Rawlsian Max-Min notion of fairness, the goal is to maximize the benefits of the least-advantaged members of society. In practice, this results in an algorithm that compares the worst possible outcome under each alternative, and chooses the one which maximizes the utility of the worst outcome, see equation 1.

$$\mathcal{J}(\theta, \lambda) := \min_{\theta} \max_{\lambda} L(\theta, \lambda) = \min_{\theta} \max_{\lambda} \sum_{i=0}^{n} \lambda_{s_i} \ell(h(x_i), y_i) \tag{1}$$

$h(.)$: a model that minimizes the loss over the training data $\mathcal{D} = 1, ..., n$,
$\ell(.)$: some loss function
$\lambda$: a learned assignment of weights that maximizes the weighted loss of some group $s$.

**Computational-Identifiability**    Multiple papers [22, 23, 24] address intersectional fairness by optimizing for group fairness between all computationally-identifiable groups in the input space. The perspective of learning over computationally-identifiable groups is similar. However, they differ from Lahoti et al. in that they assume the protected group features to be available in their input space and that they aim to minimize the gap in utility across groups via regularization.

---

[1]The TensorFlow implementation can be found on `https://github.com/google-research/google-research/tree/master/group_agnostic_fairness`

**Fairness without demographics**    Other research that focuses on fairness without demographics includes a paper by Hashimoto et al. called "fairness without demographics in repeated loss minimization" [25]. In this paper the same assumption is made that there is no full demographic information available (i.e. protected labels). First, they show that Emperical Risk Minimization (ERM) amplifies disparity over time and therefore often leads to unfair outcomes. Then, they propose a solution to this problem: Distributionally Robust Optimization (DRO).

DRO considers any worst-case distribution exceeding a given size $\alpha$ as a potential protected group and upweights them if they suffer from high loss. However, implied minorities from DRO might differ from demographic groups who are known to suffer from historical and societal biases. Focusing on improving any worst-case distributions runs the risk of focusing the optimization on noisy outliers. Therefore the authors claim DRO should not be used in high-stakes fairness applications like loans, criminality, or admissions [25]. ARL extends this by focusing on addressing computationally-identifiable errors such that fairness is improved for the unobserved groups [1].

# 3   Method

In this project we aim to reproduce, replicate, and evaluate the results obtained by Lahoti et al.. This section provides a short overview of the ARL model, and a detailed description of the approach for reproducing, replicating, and evaluating the results. Beforehand, let us clarify the notions of reproducibility and replicability used in this project.

**Reproducibility**   The same results can be obtained by different researchers repeating the experiment, making use of the same experimental setup used for generating the original results [26].

**Replicability**   The same results can be obtained when conducting experiments that follow the original method, without using the original experimental setup [26].

In this project, the reproducibility is tested using the TensorFlow implementation of the ARL model made available by Lahoti et al.. More specifically, the results as reported in the paper are compared to the results generated by the TensorFlow implementation, without making significant adjustments to the code.

The replicability is tested by re-implementing the ARL model in PyTorch based on the description of the ARL model in the paper. The provided source code is used as an example. If the experiment is replicable, the results generated using the re-implemented approach should match the results presented in Lahoti et al..

The performance of the model is evaluated on three datasets used for the prediction of: (1) adult income [27], (2) law school admission [28], and (3) criminal recidivism [29]. For detailed information about the datasets see the Appendix A.

## 3.1   Adversarially Reweighted Learning

The Adversarially Reweighted Learning (ARL) model proposed by Lahoti et al. combines the above mentioned concepts of Rawlsian fairness, fairness without demographics, and Computational-Identifiability. In this paragraph we will give a short overview of the ARL model. For a detailed description of the ARL model the reader is referred to the paper by Lahoti et al.[19].

ARL consists of two networks, a *Learner* and an *Adversary*, which are both simple feed-forward networks. A binary classification task is formulated, wherein each individual in the dataset belongs to an unobserved protected subgroup $S$, for example race and gender. The goal of the model is to maximize the minimum utility $U$ across all groups $s \in S$. However, protected group membership $S$ are not available at training or inference time. The Learner is a simple iterative learning algorithm for classification tasks that tries to minimize the expected loss at each training step. The Adversary attempts to identify the computationally-identifiable subgroups for which the Learner makes significant errors and then maximizes the weighted loss in those areas. This way, the Learner is forced to improve for those groups. Both networks are trained alternately.

The metric used for evaluating the model is the Area Under the ROC Curve (AUC). Here, ROC refers to the Receiver Operating Characteristic (ROC) curve. As such, it reflects the trade-off between correctly classified minority and majority instances. It is important to note that the ROC does not depend on the class distributions, yielding an unbiased metric of performance. Using accuracy as an evaluation metric would present a bias towards the largest groups in the data, since correctly predicting for the largest groups would contribute most to the overall performance.

### 3.2 Reproducibility

The reproducibility of the results presented by Lahoti et al. was tested on their openly available implementation on GitHub [2]. To generate the results, the instructions for installing the required packages, pre-processing the data, and training the model were followed closely. In addition to following the instructions provided by Lahoti et al., some steps were required for correctly pre-processing the data and training the model.

#### 3.2.1 Pre-processing

Lahoti et al. provided instructions for downloading and pre-processing the datasets and clear documentation on the required packages for the model training part. However, this did not include some packages required for pre-processing the data. Therefore, there was ambiguity about the correct version of these missing packages.

To prepare the three datasets used for evaluation, three *Jupyter notebook* files were provided. However, some packages used in these notebooks were not included in the documented package requirements. Therefore, the following packages were installed additionally: *Pandas*, *Seaboard*, *Sci-kit learn* and *absl.testing*. Because it was unknown which versions of these packages were needed, the latest versions were used. This, in turn, presented some deprecation issues relating to one of the functions used in the notebook. Therefore, in all the notebook files the function `reindex_axis` needed to be changed to `reindex`.

Underneath, we list the additional changes required for the individual datasets:

**COMPAS**  To prepare the COMPAS data, no additional changes other than the ones listed above were needed.

**UCI Adult**  The adult dataset contained some defects that resulted in the incorrect preparation of the dataset. The train and test files contained spaces and at the end of each entry of the income column in the test file a period was present. Moreover, the files contained a header that was incompatible with the data loader.

**LSAC**  To prepare the Law School Admissions Council dataset, the original *SAS* files were converted to *CSV* files. No explanation was provided on how to do this, so we loaded the *SAS* file as a pandas dataframe directly. Moreover, the order of columns was incorrect and had to be renamed. Later in the notebook, only the renamed columns are selected. This led to a lot of important features being discarded (including the target value). We changed this such that all features were selected.

#### 3.2.2 Training

Running the TensorFlow code using the default hyperparameters did not yield the same results as presented in the paper by Lahoti et al.. Because of this, we were unsure if these hyperparameters were the ones that were used for generating the original results, but we suspect this is not the case due to the big difference in performance. Since Lahoti et al. did not include the code used for their hyperparameter tuning, their exact optimal hyperparameters remain unknown.

The paper mentioned that the hyperparameters were chosen via a grid-search by performing 5-fold cross-validation with best overall AUC as optimization criterion. However, the corresponding code was not found on their GitHub. The only hyperparameters found in the code were the default batch size, Learner and Adversary learning rate, valued 256, 0.01, and 0.01 respectively. In addition, the amount of training steps for the model was not specified. Finally, the paper mentioned that the results were reported as an average over ten runs. However, the provided implementation only performed a single run. Additionally, the runs were not seeded, meaning that the reproduced runs are not randomized identically to the original runs. We contacted the authors about these issues by email, but did not receive a response.

Running the TensorFlow implementation with the default parameters and the default 20 training steps yielded subpar performance. Therefore, in addition to averaging the results over 10 runs and using the default parameters, the parameters found in the grid-search for our PyTorch implementation were used (see Table 1).

Some differences between TensorFlow code and model description in the article were noticed. The most important differences will be described here. For a complete overview of all differences we refer to Table **??** in the Appendix. According to the model description in the report, the Learner has two hidden layers, followed by a Softmax activation function. In the TensorFlow implementation this is a Sigmoid function. Since the classification task is binary, we used a Sigmoid in our PyTorch implementation as well. Moreover, a number of pretrain steps is defined in the TensorFlow code of the ARL. During these pretrain steps the Learner is trained without the Adversary. This step is not described in the paper.

---

[2] https://github.com/google-research/google-research/tree/master/group_agnostic_fairness

### 3.3 Replicability

Based on the article by Lahoti et al., the ARL model is replicated by implementing it in PyTorch[3]. The starting point for this implementation was the article. However, the article did not provide sufficient information for replicability. As mentioned, there were some unclarities about the model in the paper. Therefore the TensorFlow implementation was also used as guidance. This made it possible to fill in the missing information and build the ARL model as similar as possible to the original authors.

The PyTorch implementation of the ARL model follows the structure presented by Lahoti et al., with some minor changes introduced by the differences in their TensorFlow implementation. The Learner is a fully connected two layer feed-forward network with 64 and 32 hidden units in the hidden layers and a sigmoid activation function. The Adversary was implemented with one linear layer (32 hidden units) and a sigmoid activation function. In accordance with the TensorFlow implementation, both models used categorical embeddings of size 32. Moreover, the Learner and Adversary models are each optimized using a different Adagrad optimizer.

Since the optimal hyperparameters used for the results presented by Lahoti et al. were not mentioned, we performed a grid-search over an exhaustive parameter space given by batch size (16, 32, 64, 128, 256), Learner learning rate (0.001, 0.01, 0.1, 1), and Adversary learning rate (0.001, 0.01, 0.1, 1). The grid-search was performed using 5-fold cross validation on the training data. The results of this optimization are presented in Table 1. As stated before, it is unknown if the hyperparameters presented in Table 1 are the same hyperparameters used by Lahoti et al..

A difficulty for replicating the paper is the use of the term *subgroups*. In the description of the TensorFlow code *subgroups* refers to four groups (*black male, white male, black female* and *white female*). However, in the report *subgroups* seems to refer to eight subgroups; the groups as stated above, and also *male, female, black* and *white*. We chose to include all eight subgroups in our implementation.

| dataset | batch size | Learner lr | Adversary lr | steps |
|---------|-----------|-----------|-------------|-------|
| Adult   | 256       | 0.01      | 1           | 990   |
| LSAC    | 256       | 0.1       | 0.01        | 990   |
| COMPAS  | 32        | 0.01      | 1           | 470   |

Table 1: The optimal batch size, Learner learning rate, Adversary learning rate, and amount of steps found using 5-fold cross validation on the training data. The optimization was done on the PyTorch implementation.

The fairness of the model was evaluated using AUC scores. In addition to an average AUC score of the entire dataset, AUC scores were calculated for each protected group, and their intersections. This resulted in AUC scores for eight subgroups. From these subgroups, the minimum AUC score will be reported (AUC(min)), as well as the AUC score for the subgroup that is least represented in the data (AUC(minority)). Additionally, the macro-avg AUC is reported, which is the average of the AUCs of each subgroup. The scores are reported as an average over 10 runs, including the standard deviation to indicate the variability of the performance.

### 3.4 Significance evaluation

To determine if the improvement of the ARL model with respect to the baseline model is statistically significant, a significance test was added to the implementation. As a significance metric, the method by J. A. Hanley and B. J. McNeil [30, 31] was used to calculate the P-value between AUC scores of two models. A P-value below 0.05 ($P < .05$) indicates that the difference between two AUC scores is significant. A higher P-value indicates the difference is not significant [32]. Note that since the P-value cannot exceed 1, it is customary to report it without 0 before the decimal point (e.g. $P = .031$). Calculating the P-value between AUC scores requires access to the models' number of true positives and true negatives. Since we did not know these numbers for the experiments by Lahoti et al., and since their results were not reproducible, we could only conduct a significance test on our own results.

Therefore, the significance tests in our results refer to the significance of our baseline and ARL implementations in PyTorch, in stead of the provided TensorFlow implementation of Lahoti et al..

---

[3]The implementation is openly available on GitHub.

# 4 Results

This section provides a brief overview of the results. First, the results for reproducing the Lahoti et al. are presented. Second, the outcomes of our replicated PyTorch implementation are shown. Finally, the results for the significance test are presented.

## 4.1 Reproducibility

The results generated using the TensorFlow implementation of Lahoti et al. can be found in Table 2. Note that the default parameters have been used for these runs, as there were no optimal hyperparameters available. This results in lower AUC scores than Lahoti et al. report in their paper. The average AUC for the Adult dataset for Lahoti et al. is 0.907 whereas our run yields an AUC of 0.497. The AUC for the LSAC dataset of Lahoti et al. is 0.823 whereas our result is 0.518. The COMPAS dataset has an AUC of 0.748 for Lahoti et al. as opposed to 0.536 for us.

Using the optimal parameters found by our own hyperparameters search, the AUC results of the TensorFlow implemenation increases. However, the AUC scores are still lower than presented in the paper.

| dataset | AUC default | AUC optimal | AUC Lahoti et al. |
|---|---|---|---|
| Adult | 0.497 | 0.893 | 0.907 |
| LSAC | 0.518 | 0.773 | 0.823 |
| COMPAS | 0.536 | 0.754 | 0.743 |

Table 2: Reproducibility: the AUC scores for the TensorFlow ARL model by Lahoti et al. with the default parameters (only the number of training steps is altered). Left two columns are the results of one training seed, the right two columns the average over 10 different seeds.

| dataset | framework | AUC avg | AUC macro-avg | AUC min | AUC minority |
|---|---|---|---|---|---|
| Adult | Baseline Lahoti et al. | 0.898 | 0.891 | 0.867 | 0.875 |
| Adult | ARL Lahoti et al. | 0.907 | 0.915 | 0.881 | 0.942 |
| Adult | ARL TensorFlow | 0.893 | - | - | - |
| Adult | ARL PyTorch | 0.904 ± 0.0020 | 0.914 ± 0.0017 | 0.878 ± 0.0021 | 0.949 ± 0.0038 |
| LSAC | Baseline Lahoti et al. | 0.813 | 0.813 | 0.790 | 0.824 |
| LSAC | ARL Lahoti et al. | 0.823 | 0.820 | 0.798 | 0.832 |
| LSAC | ARL TensorFlow | 0.773 | - | - | - |
| LSAC | ARL PyTorch | 0.820 ± 0.0091 | 0.817 ± 0.0104 | 0.795 ± 0.0112 | 0.829 ± 0.0125 |
| COMPAS | Baseline Lahoti et al. | 0.748 | 0.730 | 0.674 | 0.774 |
| COMPAS | ARL Lahoti et al. | 0.743 | 0.727 | 0.658 | 0.785 |
| COMPAS | ARL TensorFlow | 0.754 | - | - | - |
| COMPAS | ARL PyTorch | 0.721 ± 0.0065 | 0.702 ± 0.0091 | 0.616 ± 0.0282 | 0.754 ± 0.0150 |

Table 3: Replicability: this table shows the original AUC scores presented by Lahoti et al., the AUC scores generated using the TensorFlow ARL model by Lahoti et al., and the AUC scores generated using the replicated PyTorch ARL model. The TensorFlow and PyTorch model were ran using the optimal hyperparameters described in Table 1. For the ARL PyTorch values, the standard deviation is added (mean ± std). ARL macro-avg, min and minority are absent for TensorFlow row because they were not present in the provided source code.

## 4.2 Replicability

Table 3 shows the AUC scores on the different datasets for the TensorFlow implementation with the optimal hyperparameters that resulted from the grid-search. It also shows the re-implemented ARL model in PyTorch with the same optimal hyperparameters, and the AUC scores reported in the paper by Lahoti et al.. The TensorFlow entries in Table 3 for the additional AUC metrics, i.e. AUC(macro-avg), AUC(min), and AUC(minority), are left blank because these were not reproducible from the source code.

For the Adult and LSAC datasets, the PyTorch implementations yields AUC(avg) results very close to the results of the ARL model as presented by Lahoti et al. For the COMPAS dataset this difference is slightly larger, a difference of 2 pp for AUC(avg).

Also, for the other AUC metrics (AUC(macro-avg), AUC(min) and AUC(minority)) the results of our PyTorch implementation are close to the results presented by Lahoti et al.. As was concluded by Lahoti et al., the ARL model increases the AUC for the Adult and LSAC datasets, but not for the COMPAS dataset. All AUC metrics of the COMPAS dataset are lower than the baseline model presented by Lahoti et al.. As the results show, the difference between baseline and ARL is even larger for the PyTorch implementation, in comparison to the original ARL model. For instance, AUC(min) of the PyTorch implementation is 6 pp lower than the AUC(min) of the baseline, whereas it was 2 pp (percent-point) lower for the ARL by Lahoti et al.. The largest improvement in AUC for a minority group (AUC(minority)) is for the adult dataset, an increase from 0.875 (baseline) to 0.949 (PyTorch ARL).

### 4.3 Significance evaluation

The results of the significance test are shown in Table 4. None of the P-values are $P < .05$. As such, according to the significance method [31] the performance improvement of the PyTorch implemented ARL model with respect to a simple baseline model, is not significant. It should be noted that this significance test refers only to our implemented models, and not the TensorFlow code provided by Lahoti et al..

|  | Adult | LSAC | COMPAS |
|---|---|---|---|
| Baseline (AUC) | 0.904 | 0.818 | 0.721 |
| ARL (AUC) | 0.904 | 0.829 | 0.721 |
| P-value | .963 | .977 | .977 |

Table 4: P-values calculated for ARL model and baseline.

## 5 Discussion

Reviewing this research can be divided into a discussion about (1) the reproducibility and replicability of this paper, and (2) the soundness of the research and the contribution to the field of fair AI.

### 5.1 Reproducibility and replicability

Many scientific research proves to be not reproducible and not replicable [33]. Having one-shot communication with the original authors often improves the reproducibility and replicability significantly. This was not the case in our report, as the authors did not respond to our email.

Lahoti et al. provided a thorough explanation of the model implementation in their paper. In addition, they have made a valuable contribution by making their code publicly available on GitHub. Despite this, true reproducibility was not possible. Since important information was missing, the provided source code did not yield the same results as the paper. This includes a method to prepare the data, the hyperparameters, and the amount of training steps used to train the model. As such, the provided source code did not faithfully capture several points discussed in the paper.

On the topic of replicability, we managed to implement a working PyTorch implementation of the ARL model. As such, the research was indeed replicable, albeit making a fair amount of assumptions about issues that did not become clear from the paper nor the original code.

### 5.2 Soundness and contribution

To reliably evaluate a method, it is important that the used data is sound and reflects the real world. For this research there are a few remarks to be made about the datasets in combination with the proposed model. Firstly, the used datasets are rather small. When drawing conclusions about the effectiveness of the model this should be kept in mind, since training models on small datasets often give rise to high variance in the results [34].

Another remark to make, is that the performance on the protected groups for the baseline model already displays a high performance on all three datasets (see Table 3). The ARL model does not seem to improve on this tremendously. This gives rise to the question whether applying the ARL model actually makes a significant difference. The findings presented in this report suggest that the ARL does not improve AUC significantly.

Additionally, this research uses a different definition of fairness from what most papers deploy. This definition is more beneficial for disadvantaged groups, i.e. it reduces discrimination in ML decision-making-models for worst-off groups. However, this means that people that are normally better off, may be put at a disadvantage respectively.

Despite these remarks ARL contributes to the field of ML fairness, because it provides a novel view on fairness without demographics. Whereas previous methods propose pre-defining the minimum size of the minority groups, ARL is robust to group size. Moreover, Lahoti et al. make way for future research oriented at Max-Min Fairness through their contributions in Adversarial learning.

## 6 Conclusion

In this research the paper of Lahoti et al. [1] was studied with the following objectives: (1) to test the reproducibility of the experiments by Lahoti et al., (2) to test the replicability, for which the ARL model was re-implemented in PyTorch, and lastly (3) to evaluate the significance of the reported results.

The paper "Fairness without Demographics through Adversarially Reweighted Learning" proved to be irreproducible. Since the optimal hyperparameters were not specified, the open-source TensorFlow implementation yielded lower AUC than presented in the paper.

However, after preparing the datasets, re-implementing the model in PyTorch, and conducting optimal hyperparameter search, the results of the original paper proved to be replicable. For the Adult and LSAC datasets, the results of the PyTorch implementation are very similar to the results of Lahoti et al.. For the COMPAS dataset this difference is larger. This larger difference could be attributed to the relatively small size of the COMPAS dataset, which can lead to high variance.

According to the significance tests, the difference in performance between the baseline model and the ARL model is not significant. Since the baseline performance on the dataset already yields high AUC values, it could be useful to evaluate how the model performs on datasets that do not have a high baseline AUC. Moreover, instead of maximizing the performance of the worst off-group, it could be investigated if the use of Adversarial learning is beneficial for other fairness criteria such as a utilitarian criterion, or Pareto Efficiency. Since Lahoti et al. attribute the ill-performance on the COMPAS dataset to noise, the influence of noise could additionally be evaluated. Further research into these matters could clarify if the use of Adversarially Learning is a sufficient method for improving fairness without demographics.

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

# Appendix

## Appendix A

| Dataset | Size | No. Features | Protected features | Protected groups | Prediction task |
|---|---|---|---|---|---|
| Adult | 48842 | 15 | Race, Sex | {White, Black} x {Male, Female} | Income above 50k? |
| LSAC | 26551 | 12 | Race, Sex | {White, Black} x {Male, Female} | Pass bar exam? |
| COMPAS | 7214 | 11 | Race, Sex | {White, Black} x {Male, Female} | Recidivate in 2 years? |

Table 5: Overview of the used datasets, including the size, number of features, the protected features, protected groups and prediction task of each dataset.

| Dataset | Size | % Male | % Female | % White | % Black | # Male | # Female | # White | # Black |
|---|---|---|---|---|---|---|---|---|---|
| Adult | 48842 | 0.668 | 0.332 | 0.855 | 0.096 | 32650 | 16192 | 41762 | 4685 |
| LSAC | 26551 | 0.560 | 0.440 | 0.826 | 0.067 | 14873 | 11678 | 21936 | 1790 |
| COMPAS | 7214 | 0.807 | 0.193 | 0.340 | 0.512 | 5819 | 1395 | 2454 | 3696 |

Table 6: The percentage and amount of male, female, white and black individuals in the datasets Adult, LSAC and COMPAS.

| Dataset | Male | Female | White | Black |
|---|---|---|---|---|
| $> 50K$ | 0.304 | 0.109 | 0.254 | 0.120 |
| $\leq 50K$ | 0.696 | 0.891 | 0.746 | 0.879 |
| Passing | 0.803 | 0.798 | 0.828 | 0.572 |
| Failed or not attempted | 0.197 | 0.201 | 0.172 | 0.428 |
| Recid | 0.505 | 0.380 | 0.418 | 0.551 |
| Not recid | 0.495 | 0.620 | 0.582 | 0.449 |

Table 7: Overview of intersection of protected features in the datasets Adult, LSAC and COMPAS. The table displays the predicted income, successrate, and recidivism prediction for male, female, white and black individuals.

**Appendix B**

| Description | Lahoti et al. paper | Tensorflow implementation |
| --- | --- | --- |
| Number of subgroups | 8 | 4 |
| Learner module architecture | Two linear layers and a ReLu activation function | Two linear layers and a sigmoid activation function |
| Embedding size | - | 32 |
| Pretrain steps | - | 250 |
| Optimizer | - | Adagrad |
| Hyperpameters values | Optimal hyperparamters through grid-search | Default hyperpameters, but not clear if they are optimal |
| How to handle "other" race values (e.g. Asian, Hispanic) | - | "other" race values ignored |
| Hyperparameter | Hyperpameter search | Not present |
| Results | Averages over 10 runs | One run |

Table 8: Differences between the paper by Lahoti et al. and the TensorFlow implementation

