# OpenReview forum: "Reimplementing the Adversarially Reweighted Learning model by Lahoti et al. (2020) to improve fairness without demographics"
_ML_Reproducibility_Challenge/2020 — Reject_

### Official Review · AnonReviewer3 · 2021-02-26
**Reproduction and sucessful replication, but significance evaluation unclear**

**Rating:** 6
**Confidence:** 4

**Review:**

As noted in the reproducibility summary the goal of Adversarially Reweighted Learning (ARL) is to improve the fairness of a classifier for disadvantaged groups. In contrast to other approaches the group label is not available, but has to identified from the examples.

The authors use the original code to reproduce the experiments as well as reimplement the software from scratch based on the information in the original paper. Communication with the authors of the original paper was attempted, but failed. Reproduction of the experiments (using the original code) showed large differences to the results in the original paper. Replication with the reimplemented software yielded comparable results instead. Using the optimized hyperparameters from the replication improved the results of the reproduction, too. These results are discussed in the reproducibility report in detail and the missing information for full reproducibility was pointed out there.

The report is well written and clearly shows the easy and difficult parts of reproducing and replicating the original paper. Additionally, the authors analyzed if the improved accuracy obtained by ARL was significant. But here (in table 4) the AUC values for ARL are for different groups, AUC(avg) for Adult and COMPAS data sets, AUC(minority) for the LSAC data set, and those for the baseline cannot be found in tables 2 or 3. I recommend to correct table 4 and reevaluate the significance. It may be that the improvement for AUC(avg) is not significant, but the goal of ARL is a significant improvement for AUC(minority) or AUC(min).

**Familiar With The Original Paper:**

I have read the original paper

**Reproducibility Summary:**

Report has summary

---

### Official Review · AnonReviewer2 · 2021-02-28
**Review of "Reimplementing the Adversarially Reweighted Learning model by Lahoti et al. (2020) to improve fairness without demographics"**

**Rating:** 7
**Confidence:** 4

**Review:**

This work replicates the paper "Fairness without Demographics through Adversarially Reweighted Learning." The replicating authors try the original code and implement new code to verify the results of the work and find that the results generally do not hold under a significance test. This is important to thoroughly check and replicate because such fairness mechanisms may be relied upon in real-world settings.

Pros:
+ I think the testing of both a re-implementation and original code is hugely important and well done by the authors.
+ I also appreciate the use of statistical robustness checking to make sure that results hold (or in this case, might not hold). This is the strong point of the paper to me as it expands upon the original paper and checks the quality of the results.
+ It appears that the original codebase has been updated to include hyperparameters, so this work may have done its job! That being said, a note of this should likely be made in this paper to reflect the update.

Cons:
+ While I agree that the original work should have included hyperparameters and made the code able to reproducible the exact results, I'm not sure that I follow why the authors couldn't do the exact same thing as with their pytorch implementation and run the grid search per the original paper specifications. It seems that with the replicated PyTorch code, they were able to get quite close to the original results with their grid search so it seems feasible to have done the same thing with the original tensorflow code. As a result, I'm not sure it's fair to call the original paper not reproducible in such strict terms.



Typos/Style:

Overall, this work could use another pass to clean up typos/style, some of which are below.

In the paper “fairness without Demographics through Adversarially Reweighted Learning” [1], Lahoti et al. ask themain research question: “How can we train a ML model to improve fairness when we do not know the protected groupmemberships?”. --> Fairness should be capitalized in the paper title, there shouldn't be a period after the question mark in quotes.

 original TensorFlow implementation by Lahoti et al.1. --> don't need a period after the footnote and in general this list should be separated by semi-colons not periods

 approach should match the results presented in Lahoti et al.. --> two periods

of all differences we refer to Table??in the Appendix. --> Table isn't linked correctly

as tested on their openly available implementation onGitHub2.  --> footnote usually goes after the punctuation

"Many scientific research proves" -> "Much scientific research proves" or "Many scientific research papers prove"

"Note that since the P-value cannot exceed 1, it is customary to report it without 0 before the decimal point (e.g.P=.031)." --> If this is customary, the statement can likely be omitted.

**Familiar With The Original Paper:**

I have not read the original paper

**Reproducibility Summary:**

Report has summary

---

### Official Review · AnonReviewer1 · 2021-02-28
**Please check the original paper's optimal hyperparameters.**

**Rating:** 5
**Confidence:** 3

**Review:**

The optimal hyperparameter of the original paper is now available in their Github repository. Can you reproduce their original results for both the baseline and ARL. Can you provide reasons why the original paper has a bad baseline result? Is it because they were not choosing proper hyper-parameters?

**Familiar With The Original Paper:**

I have not read the original paper

**Reproducibility Summary:**

Report has summary

---

### Decision · Program_Chairs · 2021-03-31

**Decision:**

Reject

**Comment:**

Overall reviews and/or the paper content not good enough for the AC to recommend to the journal.